# Adversarial AI applied to cross-user inter-domain and intra-domain adaptation in human activity recognition using wireless signals

**Muhammad Hassan**⊚*, **Tom Kelsey**[ID]⊚, **Fahrurrozi Rahman**[ID]

School of Computer Science, University of St. Andrews, St. Andrews, United Kingdom

⊚ These authors contributed equally to this work.
* mh353@st-andrews.ac.uk

**Data Availability Statement:** The data that supports the findings of this study is openly available in open access at: https://github.com/parisafm/CSI-HAR-Dataset.

## Abstract

In recent years, researchers have successfully recognised human activities using commercially available WiFi (Wireless Fidelity) devices. The channel state information (CSI) can be gathered at the access point with the help of a network interface controller (NIC card). These CSI streams are sensitive to human body motions and produce abrupt changes (fluctuations) in their magnitude and phase values when a moving object interacts with a transmitter and receiver pair. This sensing methodology is gaining popularity compared to traditional approaches involving wearable technology, as it is a contactless sensing strategy with no cumbersome sensing equipments fitted on the target with preserved privacy since no personal information of the subject is collected. In previous investigations, internal validation statistics have been promising. However, external validation results have been poor, due to model application to varying subjects with remarkably different environments. To address this problem, we propose an adversarial Artificial Intelligence AI model that learns and utilises domain-invariant features. We analyse model results in terms of suitability for inter-domain and intra-domain alignment techniques, to identify which is better at robustly matching the source to target domain, and hence improve recognition accuracy in cross-user conditions for HAR using wireless signals. We evaluate our model performance on different target training data percentages to assess model reliability on data scarcity. After extensive evaluation, our architecture shows improved predictive performance across target training data proportions when compared to a non-adversarial model for nine cross-user conditions with comparatively less simulation time. We conclude that inter-domain alignment is preferable for HAR applications using wireless signals, and confirm that the dataset used is suitable for investigations of this type. Our architecture can form the basis of future studies using other datasets and/or investigating combined cross-environmental and cross-user features.

**Funding:** The author(s) received no specific funding for this work.

## Introduction

Commercial-off-the-shelf (COTS) WiFi devices were initially invented for wireless communication and local area networking using wireless networking protocols. Owing to the ubiquitous nature of WiFi technologies, there are tens of billions of devices connected together in a network. Today, we are surrounded by various types of wireless signals such as WiFi, LoRa, and LTE. Earlier research has shown that the radio signals travel through multiple paths and can be used to identify the presence, location and movement of surrounding objects, via superposition at the receiver. The pervasive nature of the radio signals and their capability to demodulate the activities of the surrounding environment open the way for a new wireless sensing technology. WiFi Sensing is hence the use of commercially available WiFi devices for carrying information about users' behavior.

The field of wireless sensing is involved with the key concepts of Multiple Input Multiple Output (MIMO) [1] and Orthogonal Frequency Division Multiplexing (OFDM). MIMO is a technology that creates multiple versions of the same signal using multiple antennas at both the source and destination. These multiple versions of the same signal are helpful to both increase the signal-to-noise-ratio and reduce signal fading, since multiple copies of same signal increase the chances of the signal arriving at the receiving end successfully [1]. MIMO in WiFi devices can supply diverse and rich data concerning how signals carry information related to the surrounding environment, which we refer to as channel state information (CSI). OFDM is a modulation technique that supports a large number of carriers, each separated from the other orthogonally. It is less susceptible to selective fading, interference, and multi path effects [2]. Modern WiFi devices with IEEE 802.11 n/ac standards utilize OFDM with MIMO systems. In OFDM, data is transmitted over multiple orthogonal sub-carriers with quite narrow bandwidth. Therefore, it suffers from flat fading but this is not very severe, while co-channel interference is also avoided to a great extent. CSI data has benefits compared to received signal strength indicator (RSSI) [3]. RSSI measures the signal power on the receiver side and associates it with the distance either from the reflected object or the transmitter. This signal strength is susceptible to multi-path fading. When the transmitted signal is emitted in the environment, it gets obstructed with the surrounding objects such as buildings, vehicles and humans, which takes multiple paths before reaching at the receiver. Different signals presume different path lengths, thus suffering from fading and delay. This results in the reduction of the received signal power.

When radio signals emerge from a COTS WiFi device and spread out in the surrounding environment, they follow a multi-path propagation which induces a pattern of channel state information (CSI) at the receiving end. As a target (object or human) performs some activity under the presence of wireless environment, it creates fluctuations which exhibit distinct characteristics due to different movements in the CSI pattern. These distinct fluctuating patterns are used to train a deep learning model to predict specific activities. Fig 1 illustrates the concept of wireless sensing along with the phasor representation of a target moving from location *A* to a new location *B* covering a distance *d*. Target activity whilst between *A* and *B* will be reflected by the dynamic movement of vectors in the I-Q plane at the receiving end. When radio waves emerge from a device, they are broadly classified into three main vectors in terms of a phasor diagram. The reflection and diffraction from static objects such as walls or furniture and line of sight (LOS) contact between a transmitter and a receiver forms a static vector. In the I-Q plane, $V_s$ (in blue) is the static vector, whose length represents the magnitude and angle from I-axis to Vs is its phase value. The direct reflection from the target forms a dynamic vector. As the target moves, it causes changes in the magnitude and phase of the dynamic vector. In the same I-Q plane, $V_d$ (in red) is the

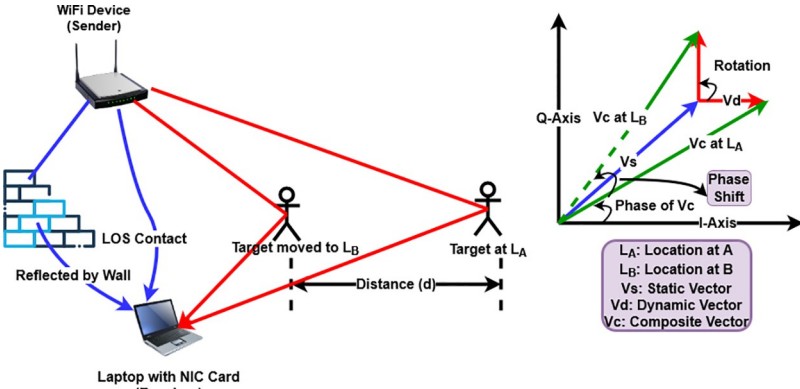

**Fig 1.** Left: Concept of wireless sensing. Right: Phasor representation.

dynamic vector appearing for two different target positions at *A* and *B*. The vector length represents its magnitude and angles from I-axis to $V_d$ at location *A* and location *B* are its phase values for these two locations. Since this vector is dynamic in nature, the phase and magnitude differences between the dynamic vectors at location *A* and *B* can be used to track the target movement. The summation of a static and dynamic vector forms a composite vector [4].

Since the fluctuations in CSI data are dependent upon surrounding objects and in fact the target characteristics can severely affects the model performance, its a challenging task to generalize a model for different cross-user conditions. Hence the work described of this study is the proposal of an adversarial model and detailed evidence to support the use of such models in this context. Our key contributions are:

- We apply inter-domain and intra-domain adaptation on an adversarial model for nine cross-user conditions using a publicly available Wi-Fi data. We achieve this by using mean discrepancy loss (MMD) and local mean discrepancy loss (LMMD).

- We evaluate the proposed model performance on different target training data proportions and show that the model is less susceptible with reduced target training data samples.

- Model average F1-micro score for nine different cross-user conditions with varying target training data proportions is 68.53% with MMD loss and 66.58% with LMMD loss.

- Model average F1-macro score for nine different cross-user conditions with varying target training data proportions is 64.28% with MMD loss and 62.6% with LMMD loss.

- Model average simulation time for nine different cross-user conditions with varying target training data proportions is not more than two to three minutes which indicates that it's a lightweight model with simple model configuration.

## Related work

The field of human activity recognition has gained popularity due to it's valuable usages in the field of activity recognition, mobile health monitoring and patient rehabilitation. The typical challenge is to concern about the model performance in cross-domain conditions such as cross-user (a classifier is trained on known users and tested on some unknown

users), cross-environment (a classifier is trained on a seen environment and tested on some unseen environment) and a combination of both of them. Models proposed by the researchers in the past performed well when they were tested on the same conditions which were used during the model training. Unfortunately, their performance suffers from acute degradation when they are tested on different environments and subjects other than those used for the model training. The activity patterns for new users and environments differ from those in the training data which makes the model less efficient in predicting activities in cross-domain conditions. Additionally, training a classifier for unseen users and environments is time consuming which also takes high computational resources [5]. Domain adaptation [6], a sub-field of transfer learning [7], is considered to be an appropriate solution for adjusting a model's parameters (weights and biases) to transfer them from one domain, refer to as source domain, to another as target domain whereas both the domains consist of domain variant features (source and target features are different from each other). Researchers in recent years resorted to unsupervised domain adaptation (UDA) [8] where adversarial learning approaches are applied to transfer domain independent features from source domain with labelled data to match with the target domain features, however, this new unseen target domain has unlabelled data samples. Virtual sample generation via geometric modelling [9], is the representation of drafting a translation function between source and target configurations. Translation function is a mathematical modelling to generate virtual samples for target movements in different locations and orientations, thus saves time to collect new training data for user's new locations and orientations. However, this method is not very effective all the time because of it's initial essential parameters estimation requirements such as users' moving speed and directions in both the configurations and their initial locations and orientations etc. Signals reflected by static objects in a specific environment are considered to be domain dependent features. These components are removed through user's motion and velocity profile modelling across different domains so that the dynamic components of the target movement can be retrieved. These dynamic components are domain invariant features as velocity profiles of different users show unique kinetic characteristics which cannot be changed with cross-environmental conditions. Also, users' velocity profiles of movements are different for different users. $v = (f\lambda)/2$ is the relation built between a user's velocity and frequency of movement that can estimate the velocity changes during the target movement [10, 11]. Transfer learning [12–14], is a way to use transferable knowledge of one domain already trained on a specific training condition (known user and environment) to train a new domain with few data samples which saves computational cost. There are two types of transfer learning as parameter transfer and feature-representation transfer. In parameter transfer [15], pre-trained models are used to fine-tune new testing domain without the need of training the entire network from scratch. These re-trained models are used to fix initial learned parameters of new domain as these layers are responsible to generate features only focused on model abstraction. They can not contribute to the model final output. A few samples from new testing domain are used to fine-tune only particular layers of the network. In feature representation [13, 14], a shared space is created between the extracted features of training and testing domain by mitigating the distinct features between them. Domain Adversarial Neural Network (DANN) [8], a type of feature representation, is one of the pioneers in the field of domain adaptation that has been applied to many of the cross-domain deep learning problems including device-free WiFi sensing. Its training works in an adversarial fashion to mismatch a generator and a domain discriminator. The generator converges to its optimal performance when discriminator fails to predict domain labels. EI [16] made the use of DANN [8] architecture to extract subject and environment independent features. They worked on three constraints to make the model effective and tolerant against

over-fitting. Confidence Control Constraint is responsible to avoid the model getting stuck on local optimum. Smoothing Constraint saves the model to be significantly different in it's predictions on neighbouring samples. Balance Constraint comes into play when model tends to assign same labels to different but similar type of activities. They changed different source domains and showed in all cases their model accuracy is higher than baseline models (VADA [17], RF [18]). Few-shot learning, is a classification problem of identifying the similarity and differences between training and testing domains using a very few labelled samples from training data. Fidora [19] is a Wireless-based localization system which can locate an objects' location fingerprints without being subject to WiFi fingerprint inconsistency such as body shapes of new users, objects in the background and daily changes in the environment. Synthetic data fingerprints are generated from labelled data fingerprints and a data augmenter (Variational Auto-Encoder) is applied for this purpose. [20]. The precedence of VAE's over traditional Auto-Encoders is their capability to generate augmented data samples from a Gaussian distribution $N(0, \gamma kI)$ of original data fingerprints. Baseline models considered in the original paper are AutoFi [21], VAE-only, and FiDo [22] which were tested on cross-user and cross-environmental conditions against Fidora [19]. Evaluation results show its average F1 score is 17.8% and 23.1% better than the benchmark in unlabeled user and varied environment respectively. WiGR [23] is a lightweight few-shot learning based gesture recognition system using WiFi devices. Network ability is its transferable domain shifting learning in new domains. Few- shot learning [24, 25] uses supervised learning to generalize a model for new tasks using only a few data samples. Model was tested against WiGeR [26], WiCatch [27], SignFi [28] and Siamese-LSTM [29] for cross- user, cross-environment and cross-location evaluations. It outperformed all of these conditions against the baseline models. They also analyzed the model complexity in terms of model's parameters and calculation required. It outperforms other few-shot learning models in model complexity such as [29–32]. JADA [33] is an unsupervised domain adaptation scheme which is proposed to tackle with the vulnerability of spatial dynamics. Evaluation results show that the model achieves 87.8% and 90.3% average recognition accuracy in cross-environmental conditions between large and small conference rooms respectively. Model is also outperforming to 2 state-of-the-art adversarial methods (DIFA [34] and ADDA [35]) under spatial dynamics. CrossGR [36] is a low cost cross-target gesture recognition model which uses generative adversarial network (GAN) for generating synthetic data samples from a small set of real-world data collected on a specific number of users. After data augmentation, it uses those labelled and synthetic data samples for eliminating out the user-related information in order to obtain gesture related features. During the back propagation, these gesture related features help the model to be trained for recognizing new users' activities. Contrastive Supervision by considering "where" to contract is a novel approach to apply contrastive loss on a time series wearable sensor data on HAR. Their key contribution is to tackle the problem of data augmentation introduced by information loss at different depth of a neural network. By using contrastive loss on intermediate layers of a network, they pushed positive augmented invariant pairs nearby and negative pairs far apart [37]. DSAN [38] is a non-adversarial model which tries to minimize the local sub-domain discrepancies within the same class of the source and target domains using local maximum mean discrepancy (LMMD) loss. DASAN [39] is an adversarial variant of DSAN [38] which is presented to solve fault diagnosis problems in different rotatory parts of machines. It focuses on global adaptation by using a discriminator for domain alignment and LMMD loss calculation between source and target activations for sub-domain alignment. During the LMMD loss calculation, they introduced pseudolabel learning [40] for generating pseudolabels for unlabelled target data.

## Preliminaries

### Channel state information

Suppose there are $M$ *Tx* antennae and $N$ *Rx* antennae in a MIMO system. Let $H$ be a CSI matrix, or called channel fading factor matrix,

$$H = \begin{bmatrix} h_{1,1} & h_{1,2} & ..... & h_{1,M} \\ h_{2,1} & h_{2,2} & ..... & h_{2,M} \\ . & . & . & . \\ . & . & . & . \\ . & . & . & . \\ h_{N,1} & h_{N,2} & ..... & h_{N,M} \end{bmatrix}$$

Each term in $H$ is a complex value representing the magnitude and phase shift of an OFDM sub-carrier in CSI stream as [41],

$$h_{i,j}(f_k) = h_{i,j}(f_k)e^{j\angle h_{i,j}f_k}, \tag{1}$$

where $h_{i,j}(f_k)$ and $\angle h_{i,j}(f_k)$ are the magnitude and phase shift of individual OFDM sub-carrier respectively. $f_k$ is the OFDM sub-carrier central frequency.

With $H$, the transmitted and received signals can be represented as

$$B(t) = H * A(t) + n(t), \tag{2}$$

where $A(t)$ and $B(t)$ are the matrices of MIMO system transmitting and receiving antennae respectively, and $n(t)$ is the additive White Gaussian noise matrix.

CSI is effective in providing precise information of a channel state. CSI streams are generated by multiple antenna pairs of a transmitter with a receiver, working at different OFDM sub-channels. These OFDM sub-channels operate at their own frequencies. Each sub-channel is associated with CSI amplitude and phase measurements. The collected CSI information over time is 4D matrix $M_{T,C,N,M}$, where $T$ is the number of WiFi signal packets, $C$ is the number of subcarriers, and $N$ and $M$ are the number of antennae. From each packet, we can extract CSI features into a magnitude and phase vector of dimension $N * M * C$. These sub-frequency carriers make different patterns for different activities, thus forming a good foundation for human activity recognition.

### Maximum mean discrepancy (MMD) loss

The maximum mean discrepancy (MMD) loss [39] measures the global distribution discrepancy between the source mean embedding and target mean embedding in the reproducing kernel Hilbert space (RKHS) provided that the source and target probability distribution is marginal. It takes two inputs, feature representations of source and target domain generated by the classifier layers as shown in Fig 2. It can be calculated as,

$$L_{MMD}(p_s, p_q) \equiv ||E_p[\phi(x^s)] - E_q[\phi(x^t)]||_H^2 \tag{3}$$

where $p_s$ is the source marginal probability distribution, $p_q$ is the target marginal probability distribution, H is the reproducing kernel Hillbert space (RKHS) endowed with a characteristic kernel k, and $\phi(.)$ is a mapping function which maps the features into the RKHS. $\phi(.)$ is associated with characteristic kernel $k(x^s, x^t) = <\phi(x^s), \phi(x^t)>$, where $(., .)$ represents the standard inner product of vectors. According to the theoretical results in [42], the source marginal

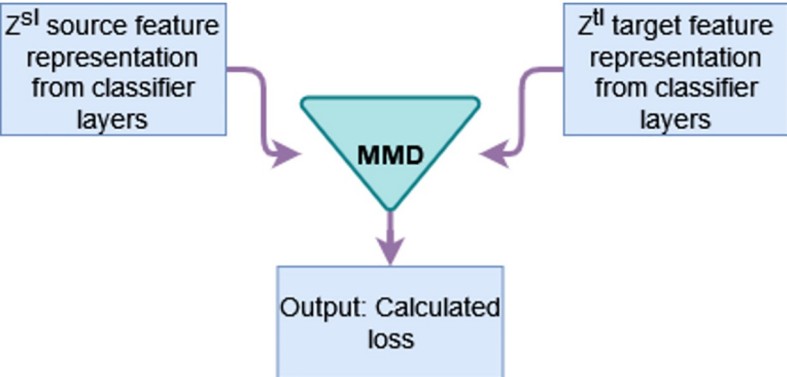

**Fig 2. MMD loss requires two inputs: $Z^{sl}$ source activation, and $Z^{tl}$ target activation.**

probability distribution is equal to the target marginal probability distribution if, and only if, $L_{MMD}(p_s, p_q) = 0$.

## Local maximum mean discrepancy (LMMD) loss

The local maximum mean discrepancy (LMMD) [38] is a variant of MDD loss, measuring the relevant sub-domains distribution discrepancies between the source mean embedding and target mean embedding in the reproducing kernel Hilbert space (RKHS). Unlike MMD loss, it focuses on the alignment of two sub-domains' relevant features within the same class of an activity. According to a particular class to which samples belong, it introduces weighted samples for each class of the activity. It takes four inputs, feature representations of source and target domain generated by the classifier layers, source true labels and target predicted labels as shown in Fig 3. Mathematically, it can be calculated as,

$$L_{LMMD}(p^{(c)}, q^{(c)}) \equiv E_c ||E_p^{(c)}[\phi(x^s)] - E_q^{(c)}[\phi(x^t)]||_H^2 \qquad (4)$$

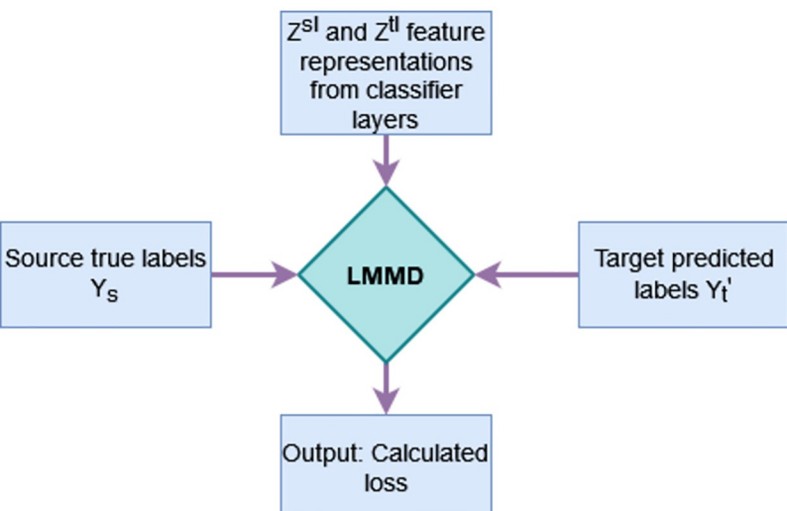

**Fig 3. LMMD loss requires four inputs: $Z^{sl}$ source activation, $Z^{tl}$ target activation, $Y_s$ source true labels and $Y_t'$ target predicted labels.**

where $p^{(c)}$ and $q^{(c)}$ are distributions of subdomains $D_s^{(c)}$ and $D_t^{(c)}$, and $x^s$ and $x^t$ are samples from source and target domains $D_s$ and $D_t$, respectively.

## Problem definition

Key challenges remain for the widespread deployment of WiFi-based sensing systems, in particular real-world environments involving users with different age, gender, height, body movement speed, location and orientation with respect to the WiFi transmitter and receiver. These aspects can severely impact the WiFi signals features and characteristics such as amplitude, phase and Doppler Frequency Shift (DFS). Consequently, if any of these factors changes from training to the testing of a model there is an inevitable degradation in the system performance caused by varying fluctuations in CSI measurements from training to the testing data samples of same activities. This creates a need to re-train the model for each new domain, requiring the extra burdens of new data collection and re-learning of model parameters and hyperparameters. Moreover, data annotation is cumbersome and time consuming because each domain carries its own specific information related to multi-path wireless propagation. Therefore, re-training a model every time for a new domain is neither feasible nor practical [15]. In order to tackle with the aforementioned problem, researchers have relied on global and sub-domain alignments on an adversarial/non-adversarial model as shown in Fig 4. These models converge easily for inter-domain alignment tasks by matching a source and target domain globally. Unfortunately, global domain adaptation neglects fine-grained information of sub-domains within the same group of different domains. Whereas, it is a time consuming process to converge these models for intra-domain alignment tasks using several loss functions. This leads to a poor transfer learning performance [38]. Cross-user transfer learning in HAR using wireless signals is a sub-domain alignment task within the same class of different activities, yet it is still unknown which type of alignment is best suitable for CSI-image based Wi-Fi data. A global alignment would be a better idea for learning domain-invariant features, by minimizing the distribution discrepancy between the source domain and target domain since CSI data for different activities appears to be quite similar without much significant domain shifting within the data. Thus, it is likely not to align perfectly on relevant sub-domain distributions. In this study we adapt an existing adversarial AI architecture in order to analyze the suitability of global and intra-class alignments for HAR domain shifting applications using wireless signals.

## Materials and methods

### Proposed method

We accessed a public dataset available at [43] on 8th June 2023. We have not had access to information that could identify individual participants during or after data collection.

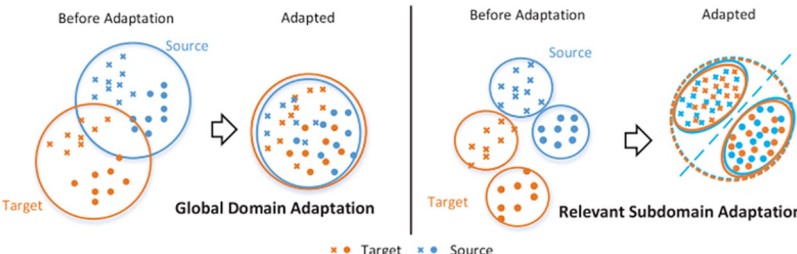

**Fig 4.** Left: domain adaptation with global alignment. Right: sub-domain adaptation with intra-class alignment [38, 39].

Available dataset has CSI magnitude values obtained from 52 sub-carriers. From these raw measurements high frequency content is filtered out as noise. Based on the nature of processed data, architecture of the feature generator can play a crucial role. Researchers have focused more on recurrent neural networks to process CSI data as a time series input with memory cells to keep track of the past inputs. This is because of the nature of CSI data which is continuous and sequential. Recurrent Neural Networks (RNNs) are supposed to be very functional in handling temporal data. These RNNs are a good option to extract key features from input CSI measurements but they need high memory requirements and their processing time is pretty long. For fully exploit the functionality of CNN with time series models in extracting shift invariant features along with the temporal information, there are plenty of 1D-CNN variants. However, these CNNs are merged with RNNs to achieve high precision but model convexity is increased thus simulation time. Our input to a 2D-CNN is a three channel RGB 64×64 CSI-image representation array of colored cells varying in intensity depending upon the magnitude values. Convolutional Neural Networks have widely been used for many applications and revolutionized the field of computer vision because of their low pre-processing requirements and remarkable results for image recognition task. Such networks can adjust filter parameters, thus useful in finding spatial and temporal dependencies in an image. ConvNets are also capable to deal with huge datasets due to their ability to reduce data dimensions. Our proposed model does not depend upon any memory cell to keep track of past inputs and its a very simple yet robust adversarial model which is suitable for applying the global and sub-domain alignments for a multi-class problem. Model is particularly chosen to investigate the impact of different alignments on cross-user domain shifting tasks using wireless sensing.

Our proposed architecture is inspired by the work presented in [39]. The main idea is to examine the effects of global as well as subdomain adaptation on HAR using device free sensing. The proposed model, Deep Adversarial Sub-Domain Adaptation (DASAN), works in three adversarial training steps. Our model architecture is shown in Fig 5 with its simulation parameters represented in Table 1. The domain shared feature extractor is a 2-D CNN. This module is responsible to extract high-level features from the raw source and target domain data samples. Since this module is shared between source and target, it maps source samples $x_s$ and target samples $x_t$ using mapping function $F_f$ with mapping parameter $\theta_f$ in such a way that $Z_s = F_f(x_s; \theta_f)$ and $Z_t = F_f(x_t; \theta_f)(Z_s, Z_t \in R^{M \times D})$ where $Z_s, Z_t$ are corresponding source and target output features with $M$ is the mini-batch size and $D$ is the feature dimensional length. Next comes a label classifier and a domain discriminator. Input to these modules is the extracted features from the previous module. The domain discriminator is responsible for predicting the corresponding domains from source and target data features. The label classifier predicts the labels' category of the extracted source and target domain features. Classifier is a mapping function $C_c$ with mapping parameter $\theta_c$ which maps the generated features to the predicted label $\hat{y}$ in such a way that $\hat{y} = C_c(Z_s, \theta_c)$. Finally, the LMMD and MMD loss functions are calculated to isolate the distribution discrepancy between the source and target activations. The LMMD loss measures the distribution discrepancy among relevant sub-domains, whereas the MMD loss measures the distribution discrepancy between the source and target distribution globally.

The label classifier is trained using the source domain labelled samples and cross entropy loss is measured between the real and predicted source labels to maximize the activity recognition accuracy on source domain that can be defined as,

$$L_{cls} = -\frac{1}{M}\left[\sum_{i=1}^{M}\sum_{c=1}^{C} I[y_{i_s} = c]log(C_c(F_f(x_{i_s}; \theta_f; \theta_c))]\right] \tag{5}$$

It also leverages pseudolabel learning for reducing the prediction uncertainty of target data

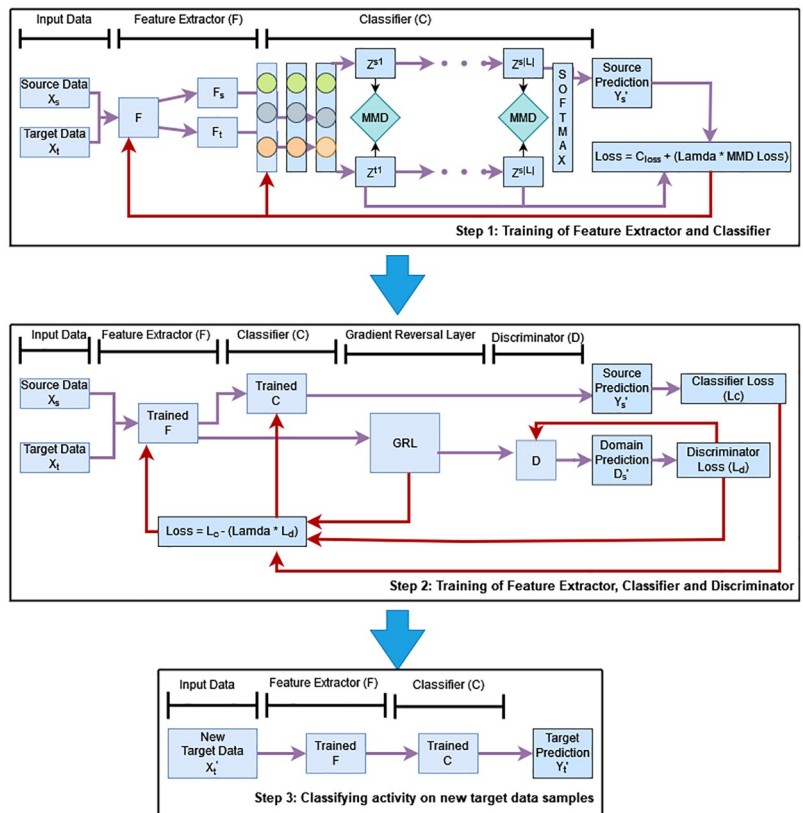

**Fig 5. Three training steps of proposed model.** The network is constructed of three modules: feature extractor, label classifier and domain discriminator. Step 1 is the training of feature extractor and classifier to obtain discriminative features. Target unlabelled samples are also used to generate pseudolabels. Step 2 is the training of feature extractor, classifier and discriminator using gradient reversal layer. Step 3 is the classification of activities on new target data samples.

unlabelled samples. Pseudolabel learning loss can be calculated as,

$$L_{Pseudo} = -\frac{1}{M}\left[\sum_{j=1}^{M}\sum_{m=1}^{C} p[\hat{y}_{j_t} = m|x_{j_t}]log(p[\hat{y}_{j_t} = m|x_{j_t})]\right] \tag{6}$$

**Table 1. Structure parameters.**

| Networks | Layers | Operations |
|---|---|---|
| Feature extractor | Conv-Pool-1 | Kernel 64-5×5, Stride 1, Padding 0; BN; ReLU; Max-Pool 3×3, Stride 2; Dropout |
| | Conv-Pool-2 | Kernel 64-5×5, Stride 1, Padding 0; BN; ReLU; Max-Pool 3×3, Stride 2; Dropout |
| | Conv-Pool-3 | Kernel 128-5×5, Stride 1, Padding 0; BN; ReLU; Max-Pool 3×3, Stride 2; Dropout |
| | Conv-Pool-4 | Kernel 256-3×3, Stride 1, Padding 0; ReLU |
| | Flatten | Nodes 256 |
| Label classifier | Linear-1 | Node 3072; ReLU |
| | Linear-2 | Node 2048; ReLU |
| | Linear-3 | Node 7; Softmax |
| Domain classifier | Linear-1 | Node 1024; ReLU |
| | Linear-2 | Node 1024; ReLU |
| | Linear-3 | Node 1; Sigmoid |

Also, the predicted labels of the label classifier for the target domain unlabelled data samples are used to calculate the LMMD and MMD losses. Thus, the objective function of label classifier can be defined as,

$$L_c = L_{cls} + \alpha L_{Pseudo} + \beta(L_{MMD}/L_{LMMD}) \tag{7}$$

where $\alpha$, and $\beta$ are the tradeoff parameters.

The purpose of domain discriminator is to minimize the global distribution discrepancy by learning domain invariant features. This adversarial role of domain discriminator is played by a two-player minmax game. The domain discriminator itself is liable to differentiate between the source and target domains as first player. The feature extractor is trained to fool the domain discriminator as second player of the game. Domain Discriminator is a mapping function $D_d$ with mapping parameter $\theta_d$ which maps the generated features in domain $d$ such as $d = D_d(f, \theta_d)(x_i \in D_s$ if $d_i = 1$ otherwise $x_j \in D_t$ if $d_j = 0$. Its adversarial loss can be defined as,

$$L_{adv} = -\frac{1}{M}\sum_{i=1}^{M} d_i log[D_d(F_f(x_{i_s}; \theta_f); \theta_d)] - \frac{1}{M}\sum_{i=1}^{M}(1 - d_i)log[D_d(F_f(x_{j_t}; \theta_f); \theta_d)] \tag{8}$$

The total loss of the model can be calculated as,

$$L_{total} = L_{cls} - \gamma L_{adv} + \beta L_{LMMD} + \alpha L_{Pseudo} \qquad \text{(in case of LMMD Loss)} \tag{9}$$

$$L_{total} = L_{cls} - \gamma L_{adv} + \beta L_{MMD} + \alpha L_{Pseudo} \qquad \text{(in case of MMD Loss)} \tag{10}$$

where $L_{cls}$ is the classifier loss, $L_{adv}$ is the discriminator adversarial loss, $\alpha$, $\beta$ and $\gamma$ are the tradeoff parameters.

## Experimental results

### Dataset

We use a public dataset available at [43] to assess model performance, named as the Parisafm dataset. The dataset was collected with the involvement of 3 volunteers, thus suitable for cross-user domain adaptation. The participants performed 7 different activities including walk, run, fall, lie down, sit down, stand up, and bend in an experimental environment. Each activity was repeated for 20 trials. In total, there are 420 labelled data samples which are equally divided among three different subjects. For adversarial training the source domain is always equipped with labeled samples for a particular subject/combination of subjects, while the target domain is treated as unlabeled data samples coming from the other subject/combination of subjects during model training. The Raspberry Pi was used as a WiFi-enabled platform for packet reception and a Nexmon Tool [44] was employed for data collection process. Each subcarrier has a complex representation of CSI values. These complex values have magnitude and phase information about a specific activity. For mode simulation, only CSI magnitude values are being employed. A low pass filter is used for the reduction of high-frequency content which is treated as noise. These values are normalized between 0 and 255 for a colored image representation. These RGB colored images are then generated as a MATLAB pseudocolor plot, shown in Fig 6. This results in an array of colored cells also known as a face. Each image is resized to 64×64 scale.

### Model evaluation

To evaluate the inter- and intra-domain adaptation on HAR using wireless sensor data comprehensively, we have two different variations of the proposed model, with their transfer

(a) Bend   (b) Fall   (c) Lie Down   (d) Run   (e) Sit Down   (f) Stand Up   (g) Walk

**Fig 6. CSI RGB images for different activities.**

results being compared to another model, Deep Subdomain Adaptation Network (DSAN) [38]. DSAN [38] is a non-adversarial model with a simple architecture of a shared feature extractor and a classifier. Features generated by the extractor for source labelled data and target unlabelled data are fed to the classifier layers one at a time. Thereafter, maximum mean discrepancy (MMD) and local maximum mean discrepancy (LMMD) losses are calculated between these source and target activations for examining the effects of global and subdomain alignments respectively. The proposed model is also tested for global and local sub-domain adaptations using the same loss minimization functions that is DASAN-LMMD and DASAN-MMD. Finally, these two variations of proposed model are compared with DSAN-MMD and DSAN-LMMD against the measuring parameters of model activity recognition micro- and macro-F1 scores, the harmonic mean of precision and recall, on cross-user domain shifting tasks. Micro-F1 score aggregates the contributions of all instances, and the macro-F1 score computes the metric independently for each class and then takes the average [45]. Since we have an imbalanced dataset, we also report macro-F1 score, which takes equal contribution from majority and minority classes to achieve objective results. Simulation time for each model is additionally measured for comparison.

The dataset used for model evaluation has three different subjects involved for performing seven different activities. We have tested each model for nine different domain shifting tasks with subject 1, 2 and 3 are interchangeably used for source to target domains. In order to report our model simulation results, we are following evaluation approach mentioned in [45]. Each case is run ten times and their average is calculated for an unbiased models comparison. We also compute and report 95% confidence intervals for each performance metric. The cross-user domain shifting task measures the accuracy of adopting an activity model trained on one user (male/female) with some physical appearance (e.g., weight, height, age) to another with different physical appearance.

**Models comparison of micro- and macro-F1 scores.** Tables 2–9, report the micro- and macro-F1 scores of DASAN and baseline technique with MMD and LMMD losses on nine cross-user experiments with different target data training samples varying from 100% to 10%. These are averaged F1 scores over 10 runs of the nine cross-user experiments reported in the table. DASAN-MMD obtains the highest average of averaged micro- and macro-F1 scores of nine cross-user domain-shifting tasks on varying target data training samples: 0.69 and 0.64, which is 0.019 and 0.017 higher than DASAN-LMMD, the second best performing technique. In addition, DASAN-MMD outperforms DSAN-MMD with 0.094 and 0.105 in micro- and macro-F1 scores, whereas it is 0.118 and 0.14 higher in micro- and macro-F1 scores than DSAN-LMMD, the least performing technique among all. We can also observe the DASAN-MMD model reliability with reduced target data training samples that is no less than 0.62 and 0.57 for averaged micro- and macro-F1 scores even for the worst case of only 10% of target data training samples. This concludes that global adaptation is a better option for HAR using wireless signals in terms of achieving higher model micro- and macro-F1 scores. Looking more closely at different cross-user tasks on the Parisafm dataset, we have plotted the averaged micro- and macro-F1 scores on varying target data training samples depicted in Figs 7 and 8.

**Table 2. Average micro-F1 scores of DSAN-LMMD on CSI image dataset across all training percentages.**

| Task | Deep subdomain adaptation network (DSAN) with LMMD loss | | | | | |
|---|---|---|---|---|---|---|
| | Micro-F1 score with different target data training samples percentages | | | | | |
| | 100% with (95% CI) | 80% with (95% CI) | 60% with (95% CI) | 40% with (95% CI) | 20% with (95% CI) | 10% with (95% CI) |
| (S1+S2)−>S3 | 75.00 (75.88-74.38) | 74.70 (75.54-74.04) | 75.80 (76.64-75.13) | 76.60 (77.46-75.93) | 77.30 (78.08-76.53) | 78.40 (79.02-77.45) |
| (S1+S3)−>S2 | 55.00 (55.94-54.84) | 59.80 (60.72-59.53) | 57.30 (58.60-57.46) | 58.60 (59.40-58.23) | 59.40 (60.39-59.20) | 62.20 (62.88-61.64) |
| (S2+S3)−>S1 | 58.20 (59.03-57.87) | 61.00 (61.81-60.59) | 57.20 (58.03-56.89) | 57.20 (58.16-57.02) | 57.10 (58.06-56.92) | 56.70 (57.73-56.59) |
| S1−>S2 | 53.10 (54.10-53.04) | 54.30 (55.19-54.10) | 50.00 (51.00-50.00) | 55.00 (55.90-54.80) | 58.40 (59.25-58.08) | 51.50 (52.27-51.24) |
| S1−>S3 | 53.90 (54.63-53.55) | 50.20 (50.97-49.96) | 56.50 (57.11-55.98) | 52.60 (53.49-52.44) | 54.30 (55.03-53.94) | 49.60 (50.20-49.21) |
| S2−>S1 | 49.60 (50.35-49.36) | 48.90 (49.67-48.69) | 48.20 (48.92-47.96) | 51.20 (51.93-50.91) | 50.20 (50.86-49.86) | 48.90 (49.64-48.66) |
| S2−>S3 | 66.40 (67.29-65.96) | 66.70 (67.59-66.26) | 67.90 (68.69-67.33) | 70.80 (71.60-70.18) | 67.50 (68.33-66.98) | 67.40 (68.17-66.82) |
| S3−>S1 | 48.60 (49.30-48.33) | 49.20 (49.88-48.90) | 46.40 (47.01-46.08) | 48.50 (49.16-48.19) | 48.50 (49.08-48.11) | 47.10 (47.77-46.82) |
| S3−>S2 | 48.10 (48.74-47.78) | 44.70 (45.30-44.41) | 45.80 (46.37-45.45) | 43.80 (44.31-43.44) | 46.20 (46.86-45.94) | 47.40 (48.15-47.21) |
| **Average** | **56.43** | **56.61** | **56.12** | **57.14** | **57.66** | **56.58** |

Note: S1 means subject 1, S2 means subject 2, S3 means subject 3, CI means confidence level

**Table 3. Average micro-F1 scores of DSAN-MMD on CSI image dataset across all training percentages.**

| Task | Deep subdomain adaptation network (DSAN) with MMD loss | | | | | |
|---|---|---|---|---|---|---|
| | Micro-F1 score with different target data training samples percentages | | | | | |
| | 100% with (95% CI) | 80% with (95% CI) | 60% with (95% CI) | 40% with (95% CI) | 20% with (95% CI) | 10% with (95% CI) |
| (S1+S2)−>S3 | 79.30 (80.10-78.52) | 82.80 (83.53-81.87) | 80.90 (81.68-80.06) | 81.90 (82.68-81.04) | 85.00 (85.68-83.98) | 76.30 (76.93-75.40) |
| (S1+S3)−>S2 | 65.00 (65.87-64.57) | 63.40 (64.41-63.14) | 55.80 (56.94-55.82) | 57.50 (58.37-57.22) | 61.80 (62.86-61.63) | 63.10 (63.87-62.61) |
| (S2+S3)−>S1 | 59.00 (59.79-58.61) | 60.70 (61.47-60.25) | 57.30 (58.14-57.00) | 60.80 (61.67-60.46) | 61.40 (62.17-60.94) | 59.40 (60.31-59.13) |
| S1−>S2 | 53.60 (54.58-53.51) | 45.60 (46.67-45.76) | 54.40 (55.21-54.12) | 46.10 (47.26-46.34) | 52.30 (53.20-52.15) | 40.30 (41.14-40.33) |
| S1−>S3 | 63.10 (63.71-62.45) | 66.10 (66.73-65.40) | 59.30 (59.87-58.69) | 59.00 (59.76-58.58) | 62.80 (63.42-62.17) | 51.50 (52.06-51.03) |
| S2−>S1 | 52.30 (53.02-51.98) | 52.20 (52.92-51.88) | 49.30 (49.92-48.94) | 50.90 (51.59-50.58) | 52.30 (52.90-51.86) | 54.10 (54.76-53.68) |
| S2−>S3 | 69.80 (70.59-69.19) | 71.40 (72.16-70.73) | 72.10 (72.86-71.42) | 71.70 (72.52-71.08) | 71.40 (72.16-70.73) | 74.10 (74.84-73.35) |
| S3−>S1 | 49.60 (50.30-49.30) | 50.40 (51.11-50.10) | 50.30 (50.94-49.93) | 50.20 (50.87-49.86) | 52.30 (52.92-51.87) | 47.40 (48.11-47.17) |
| S3−>S2 | 46.80 (47.49-46.55) | 47.00 (47.59-46.65) | 46.70 (47.29-46.36) | 46.40 (46.90-45.97) | 49.00 (49.73-48.75) | 49.50 (50.21-49.22) |
| **Average** | **59.83** | **59.96** | **58.46** | **58.28** | **60.92** | **57.3** |

Note: S1 means subject 1, S2 means subject 2, S3 means subject 3, CI means confidence level

**Table 4. Average micro-F1 scores of DASAN-LMMD on CSI image dataset across all training percentages.**

| Task | Deep Adversarial subdomain adaptation network (DASAN) with LMMD loss | | | | | |
|---|---|---|---|---|---|---|
| | Micro-F1 score with different target data training samples percentages | | | | | |
| | 100% with (95% CI) | 80% with (95% CI) | 60% with (95% CI) | 40% with (95% CI) | 20% with (95% CI) | 10% with (95% CI) |
| (S1+S2)−>S3 | 80.70 (81.50-79.89) | 80.90 (81.73-80.12) | 79.50 (80.31-78.72) | 81.90 (82.71-81.07) | 76.50 (77.37-75.84) | 70.70 (71.44-70.02) |
| (S1+S3)−>S2 | 74.60 (75.41-73.92) | 77.90 (78.71-77.15) | 78.10 (78.90-77.34) | 72.60 (73.40-71.94) | 78.50 (79.21-77.64) | 67.70 (68.36-67.00) |
| (S2+S3)−>S1 | 71.80 (72.60-71.16) | 70.70 (71.50-70.08) | 70.60 (71.31-69.90) | 71.10 (71.86-70.44) | 67.80 (68.60-67.25) | 69.30 (69.99-68.61) |
| S1−>S2 | 71.20 (72.06-70.64) | 72.90 (73.76-72.30) | 72.50 (73.19-71.74) | 72.10 (72.77-71.33) | 70.70 (71.41-69.99) | 63.30 (63.76-62.49) |
| S1−>S3 | 64.60 (65.28-63.99) | 65.90 (66.48-65.16) | 63.10 (63.79-62.53) | 66.80 (67.60-66.27) | 64.90 (65.70-64.40) | 56.10 (56.65-55.52) |
| S2−>S1 | 60.00 (60.52-59.32) | 62.90 (63.44-62.18) | 57.30 (57.97-56.82) | 58.80 (59.46-58.28) | 55.50 (56.06-54.95) | 57.80 (58.40-57.24) |
| S2−>S3 | 73.10 (73.85-72.39) | 76.10 (76.86-75.34) | 76.80 (77.59-76.05) | 76.40 (77.20-75.68) | 73.90 (74.69-73.22) | 74.00 (74.74-73.26) |
| S3−>S1 | 57.40 (58.04-56.89) | 55.80 (56.50-55.38) | 55.70 (56.31-55.20) | 56.00 (56.61-55.49) | 53.90 (54.53-53.46) | 53.00 (53.50-52.44) |
| S3−>S2 | 55.50 (56.05-54.94) | 52.60 (53.20-52.15) | 52.30 (52.95-51.90) | 48.50 (49.07-48.10) | 53.90 (54.37-53.29) | 53.00 (53.39-52.33) |
| **Average** | **67.66** | **68.41** | **67.32** | **67.13** | **66.18** | **62.77** |

Note: S1 means subject 1, S2 means subject 2, S3 means subject 3, CI means confidence level

**Table 5. Average micro-F1 scores of DASAN-MMD on CSI image dataset across all training percentages.**

| Task | Deep adversarial subdomain adaptation network (DASAN) with MMD loss | | | | | |
|---|---|---|---|---|---|---|
| | Micro-F1 score (%) with different target data training samples percentages | | | | | |
| | 100% with (95% CI) | 80% with (95% CI) | 60% with (95% CI) | 40% with (95% CI) | 20% with (95% CI) | 10% with (95% CI) |
| (S1+S2)−>S3 | 81.20 (82.02-80.40) | 80.10 (80.88-79.28) | 81.00 (81.85-80.23) | 81.60 (82.41-80.78) | 81.50 (82.40-80.77) | 70.60 (71.30-69.89) |
| (S1+S3)−>S2 | 79.40 (80.29-78.71) | 81.50 (82.39-80.76) | 80.10 (80.94-79.34) | 78.60 (79.49-77.91) | 79.00 (79.80-78.22) | 66.50 (67.14-65.81) |
| (S2+S3)−>S1 | 74.60 (75.41-73.91) | 74.30 (75.10-73.62) | 71.30 (72.02-70.60) | 73.50 (74.28-72.81) | 73.50 (74.34-72.87) | 70.80 (71.54-70.13) |
| S1−>S2 | 75.70 (76.53-75.01) | 78.70 (79.58-78.01) | 71.10 (71.78-70.36) | 69.90 (70.54-69.15) | 71.10 (71.82-70.40) | 55.90 (56.31-55.19) |
| S1−>S3 | 67.90 (68.66-67.30) | 64.70 (65.32-64.03) | 65.50 (66.19-64.88) | 72.80 (73.65-72.20) | 71.60 (72.43-70.99) | 55.50 (56.04-54.93) |
| S2−>S1 | 56.60 (57.11-55.98) | 57.30 (57.74-56.60) | 62.10 (62.82-61.58) | 62.80 (63.53-62.27) | 55.40 (55.95-54.84) | 58.70 (59.31-58.13) |
| S2−>S3 | 74.50 (75.28-73.79) | 76.00 (76.76-75.24) | 76.50 (77.26-75.73) | 77.40 (78.20-76.65) | 77.50 (78.35-76.80) | 72.00 (72.67-71.23) |
| S3−>S1 | 62.10 (62.83-61.59) | 57.40 (58.04-56.89) | 58.40 (59.04-57.87) | 57.90 (58.51-57.36) | 56.80 (57.42-56.28) | 53.30 (53.84-52.77) |
| S3−>S2 | 52.80 (53.26-52.21) | 54.60 (55.18-54.09) | 58.60 (59.33-58.16) | 56.70 (57.50-56.36) | 59.70 (60.42-59.23) | 58.00 (58.67-57.51) |
| **Average** | **69.42** | **70.31** | **69.4** | **70.13** | **69.57** | **62.37** |

Note: S1 means subject 1, S2 means subject 2, S3 means subject 3, CI means confidence level

**Table 6. Average macro-F1 scores of DSAN-LMMD on CSI image dataset across all training percentages.**

| Task | Deep subdomain adaptation network (DSAN) with LMMD loss | | | | | |
|---|---|---|---|---|---|---|
| | Macro-F1 score with different target data training samples percentages | | | | | |
| | 100% with (95% CI) | 80% with (95% CI) | 60% with (95% CI) | 40% with (95% CI) | 20% with (95% CI) | 10% with (95% CI) |
| (S1+S2)−>S3 | 68.20 (69.01-67.65) | 68.30 (69.15-67.79) | 70.30 (71.12-69.71) | 70.60 (71.45-70.04) | 71.00 (71.72-70.30) | 72.50 (72.99-71.54) |
| (S1+S3)−>S2 | 47.40 (48.29-47.34) | 52.00 (52.96-51.92) | 48.80 (49.91-48.93) | 50.70 (51.51-50.50) | 50.70 (51.53-50.52) | 56.10 (56.79-55.66) |
| (S2+S3)−>S1 | 55.90 (56.85-55.73) | 59.20 (60.08-58.89) | 53.10 (54.05-52.99) | 52.40 (53.40-52.35) | 49.80 (50.74-49.74) | 48.30 (49.44-48.48) |
| S1−>S2 | 46.90 (47.90-46.96) | 47.20 (48.07-47.13) | 44.80 (45.76-44.86) | 50.70 (51.58-50.57) | 51.60 (52.50-51.47) | 46.50 (47.18-46.25) |
| S1−>S3 | 50.20 (50.92-49.92) | 48.90 (49.66-48.69) | 54.60 (55.19-54.10) | 49.70 (50.53-49.54) | 51.10 (51.79-50.76) | 46.60 (47.18-46.25) |
| S2−>S1 | 44.70 (45.45-44.56) | 43.70 (44.53-43.65) | 43.60 (44.34-43.47) | 46.40 (47.11-46.18) | 45.20 (45.83-44.92) | 44.50 (45.27-44.38) |
| S2−>S3 | 53.90 (54.69-53.61) | 55.30 (56.11-55.00) | 55.40 (56.09-54.98) | 58.20 (58.91-57.75) | 55.50 (56.24-55.13) | 55.80 (56.48-55.37) |
| S3−>S1 | 43.70 (44.42-43.55) | 41.80 (42.49-41.65) | 39.00 (39.64-38.86) | 42.70 (43.33-42.47) | 42.50 (43.04-42.19) | 39.50 (40.20-39.41) |
| S3−>S2 | 37.90 (38.45-37.69) | 34.30 (34.79-34.10) | 34.80 (35.29-34.59) | 33.50 (33.95-33.28) | 36.90 (37.50-36.76) | 36.60 (37.22-36.49) |
| **Average** | **49.87** | **50.08** | **49.38** | **50.54** | **50.48** | **49.6** |

Note: S1 means subject 1, S2 means subject 2, S3 means subject 3, CI means confidence level

**Table 7. Average macro-F1 scores of DSAN-MMD on CSI image dataset across all training percentages.**

| Task | Deep subdomain aadaptation network (DSAN) with MMD loss | | | | | |
|---|---|---|---|---|---|---|
| | Macro-F1 score with different target data training samples percentages | | | | | |
| | 100% with (95% CI) | 80% with (95% CI) | 60% with (95% CI) | 40% with (95% CI) | 20% with (95% CI) | 10% with (95% CI) |
| (S1+S2)−>S3 | 76.60 (77.38-75.84) | 80.00 (80.69-79.09) | 76.50 (77.22-75.69) | 79.00 (79.74-78.16) | 82.00 (82.59-80.95) | 69.80 (70.31-68.92) |
| (S1+S3)−>S2 | 57.40 (58.21-57.06) | 57.30 (58.23-57.09) | 47.50 (48.60-47.65) | 49.90 (50.74-49.74) | 52.70 (53.66-52.60) | 56.60 (57.29-56.16) |
| (S2+S3)−>S1 | 58.60 (59.42-58.25) | 60.80 (61.67-60.45) | 54.70 (55.59-54.50) | 60.30 (61.19-59.98) | 61.60 (62.40-61.17) | 56.30 (57.24-56.11) |
| S1−>S2 | 48.10 (49.06-48.10) | 41.70 (42.77-41.93) | 49.30 (50.14-49.16) | 40.40 (41.38-40.57) | 49.60 (50.52-49.52) | 35.90 (36.65-35.93) |
| S1−>S3 | 59.50 (60.08-58.89) | 63.40 (63.99-62.72) | 57.80 (58.36-57.20) | 56.10 (56.81-55.69) | 59.50 (60.09-58.90) | 49.00 (49.54-48.56) |
| S2−>S1 | 49.50 (50.23-49.24) | 49.20 (50.00-49.01) | 45.30 (45.96-45.05) | 47.70 (48.42-47.47) | 48.60 (49.20-48.22) | 52.20 (52.86-51.82) |
| S2−>S3 | 58.00 (58.73-57.57) | 61.80 (62.51-61.27) | 63.00 (63.70-62.44) | 59.50 (60.22-59.03) | 60.30 (60.96-59.76) | 62.50 (63.12-61.87) |
| S3−>S1 | 44.30 (44.98-44.10) | 44.80 (45.46-44.56) | 45.40 (46.04-45.14) | 41.60 (42.26-41.43) | 45.80 (46.37-45.45) | 39.00 (39.68-38.90) |
| S3−>S2 | 36.60 (37.17-36.44) | 36.70 (37.18-36.45) | 36.00 (36.49-35.77) | 37.80 (38.22-37.46) | 37.60 (38.24-37.49) | 40.20 (40.79-39.99) |
| **Average** | **54.29** | **55.08** | **52.83** | **52.48** | **55.3** | **51.28** |

Note: S1 means subject 1, S2 means subject 2, S3 means subject 3, CI means confidence level

**Table 8. Average macro-F1 scores of DASAN-LMMD on CSI image dataset across all training percentages.**

| Task | Deep adversarial subdomain adaptation network (DASAN) with LMMD loss | | | | | |
|------|------|------|------|------|------|------|
| | Macro-F1 score with different target data training samples percentages | | | | | |
| | 100% with (95% CI) | 80% with (95% CI) | 60% with (95% CI) | 40% with (95% CI) | 20% with (95% CI) | 10% with (95% CI) |
| (S1+S2)−>S3 | 77.00 (77.77-76.23) | 74.90 (75.68-74.18) | 74.30 (75.07-73.58) | 76.80 (77.56-76.02) | 71.40 (72.23-70.81) | 61.90 (62.55-61.31) |
| (S1+S3)−>S2 | 69.89 (70.68-69.28) | 73.80 (74.58-73.10) | 74.20 (74.97-73.48) | 67.00 (67.76-66.42) | 73.60 (74.23-72.76) | 62.40 (63.00-61.75) |
| (S2+S3)−>S1 | 73.70 (74.55-73.07) | 73.60 (74.43-72.96) | 73.10 (73.84-72.38) | 73.10 (73.89-72.43) | 69.60 (70.46-69.06) | 71.60 (72.32-70.88) |
| S1−>S2 | 68.50 (69.33-67.96) | 69.30 (70.16-68.77) | 68.70 (69.35-67.98) | 69.00 (69.63-68.25) | 66.70 (67.37-66.03) | 59.80 (60.19-59.00) |
| S1−>S3 | 58.60 (59.23-58.06) | 61.00 (61.51-60.29) | 56.80 (57.43-56.30) | 61.80 (62.54-61.30) | 59.30 (60.05-58.86) | 50.60 (51.09-50.08) |
| S2−>S1 | 61.10 (61.61-60.39) | 63.10 (63.61-62.35) | 56.70 (57.41-56.28) | 58.40 (59.09-57.92) | 53.20 (53.74-52.67) | 59.00 (59.62-58.44) |
| S2−>S3 | 63.60 (64.27-63.00) | 66.40 (67.07-65.74) | 67.20 (67.91-66.56) | 66.80 (67.53-66.20) | 63.50 (64.21-62.94) | 62.20 (62.83-61.58) |
| S3−>S1 | 56.30 (56.95-55.82) | 53.70 (54.43-53.35) | 52.90 (53.50-52.44) | 54.00 (54.61-53.53) | 51.50 (52.15-51.12) | 53.10 (53.59-52.53) |
| S3−>S2 | 45.90 (46.35-45.43) | 42.90 (43.41-42.55) | 42.80 (43.36-42.50) | 39.40 (39.89-39.10) | 51.50 (51.92-50.89) | 53.10 (53.45-52.38) |
| **Average** | **63.84** | **64.3** | **62.97** | **62.92** | **62.26** | **59.3** |

Note: S1 means subject 1, S2 means subject 2, S3 means subject 3, CI means confidence level

**Table 9. Average macro-F1 scores of DASAN-MMD on CSI image dataset across all training percentages.**

| Task | Deep adversarial subdomain adaptation network (DASAN) with MMD loss | | | | | |
|------|------|------|------|------|------|------|
| | Macro-F1 score with different target data training samples percentages | | | | | |
| | 100% with (95% CI) | 80% with (95% CI) | 60% with (95% CI) | 40% with (95% CI) | 20% with (95% CI) | 10% with (95% CI) |
| (S1+S3)−>S2 | 61.90 (62.55-61.31) | 75.60 (76.46-74.95) | 75.40 (76.21-74.70) | 70.90 (71.76-70.34) | 69.40 (70.10-68.72) | 60.90 (61.47-60.25) |
| (S2+S3)−>S1 | 78.30 (79.15-77.58) | 77.30 (78.13-76.58) | 73.50 (74.25-72.78) | 75.90 (76.72-75.20) | 76.20 (77.08-75.56) | 71.60 (72.36-70.92) |
| S1−>S2 | 74.50 (75.33-73.84) | 76.30 (77.21-75.68) | 67.10 (67.74-66.40) | 66.10 (66.71-65.39) | 66.60 (67.28-65.94) | 51.20 (51.57-50.54) |
| S1−>S3 | 61.40 (62.12-60.89) | 56.20 (56.73-55.60) | 59.40 (60.04-58.85) | 68.20 (69.03-67.66) | 65.70 (66.48-65.17) | 49.30 (49.77-48.79) |
| S2−>S1 | 57.20 (57.71-56.56) | 57.60 (58.03-56.88) | 63.80 (64.60-63.33) | 64.40 (65.18-63.89) | 53.40 (53.93-52.86) | 60.30 (60.93-59.72) |
| S2−>S3 | 65.50 (66.20-64.89) | 66.50 (67.16-65.83) | 68.50 (69.18-67.81) | 69.80 (70.53-69.13) | 67.20 (67.99-66.65) | 62.50 (63.06-61.81) |
| S3−>S1 | 60.50 (61.24-60.03) | 65.10 (66.04-64.74) | 55.60 (56.21-55.10) | 57.00 (57.62-56.48) | 56.70 (57.33-56.20) | 51.00 (51.52-50.50) |
| S3−>S2 | 45.30 (45.67-44.76) | 46.60 (47.12-46.19) | 50.20 (50.88-49.87) | 47.90 (48.65-47.69) | 47.10 (47.74-46.80) | 45.30 (45.87-44.97) |
| **Average** | **65.93** | **66.43** | **65.47** | **66.27** | **64.31** | **57.26** |

Note: S1 means subject 1, S2 means subject 2, S3 means subject 3, CI means confidence level

**Models comparison of training time.** Tables 10–13, report the training times of DASAN and baseline technique with MMD and LMMD losses on nine cross-user experiments with different target data training samples varying from 100% to 10%. These are averaged training times over 10 runs of the nine cross-user experiments reported in the table. DASAN-MMD obtains a moderate average of averaged training times of nine cross-user domain-shifting tasks on varying target data training samples: 130.36 sec, which is 22.16 sec shorter than that of DASAN-LMMD, taking the longest training time among all. However, DASAN-MMD takes 31.22 sec more than DSAN-LMMD in model training time, whereas it takes 41.38 sec longer than DSAN-MMD, taking the shortest training time among all. There is a trade-off between higher model accuracy and shorter training time. However, DASAN-MMD still has the highest recognition accuracy with shorter training time as compared to DASAN-LMMD and moderate among all. Fig 9 shows the comparison of average training times of all the models for different target data training samples in a histogram plot.

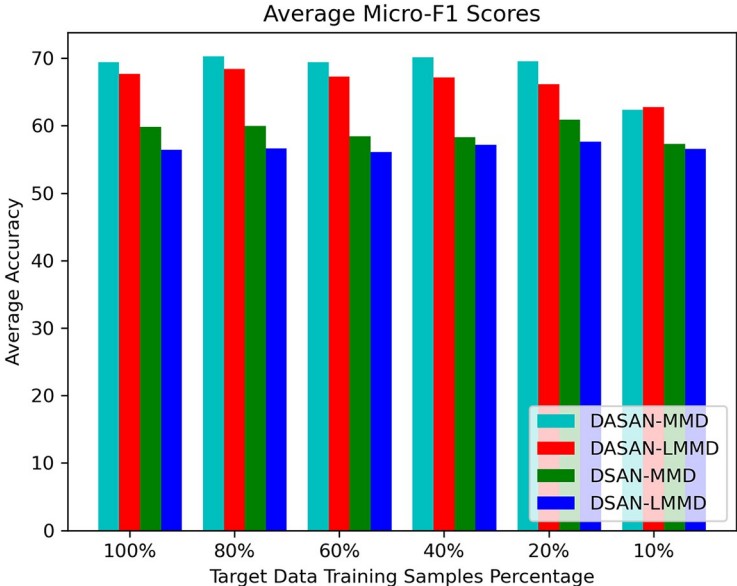

**Fig 7. Comparison of average Micro-F1 scores of DASAN-MMD, DASAN-LMMD, DSAN-MMD, & DSAN-LMMD.**

## Conclusion

In this study we both propose the use of adversarial models and supply detailed evidence to support the proposal. We have shown that our model has utility for finding the impacts of global and sub-domain adaptation on cross-user domain transferring tasks on HAR using

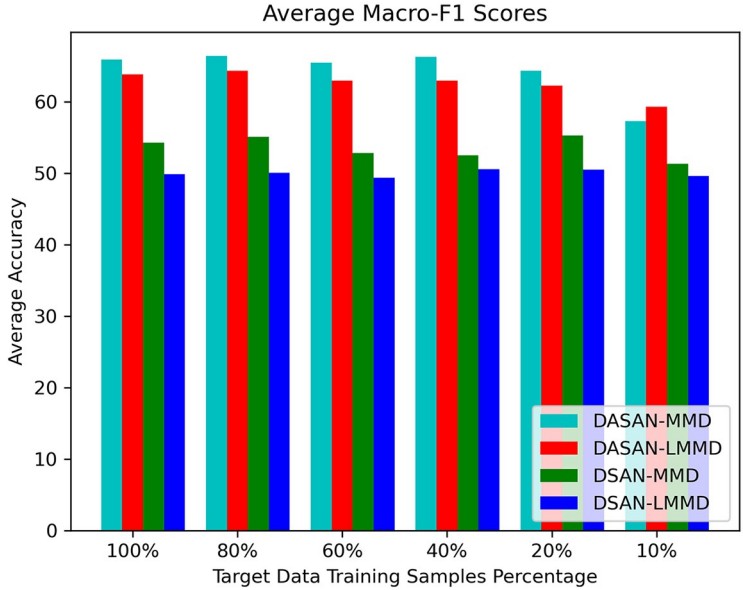

**Fig 8. Comparison of average Macro-F1 scores of DASAN-MMD, DASAN-LMMD, DSAN-MMD, & DSAN-LMMD.**

**Table 10. Average training times of DSAN-LMMD on CSI image dataset across all training percentages.**

| Task | Deep subdomain adaptation network (DSAN) with LMMD loss | | | | | |
|---|---|---|---|---|---|---|
| | Training Time in seconds with different target data training samples percentages | | | | | |
| | 100% with (95% CI) | 80% with (95% CI) | 60% with (95% CI) | 40% with (95% CI) | 20% with (95% CI) | 10% with (95% CI) |
| (S1+S2)−>S3 | 153.36 (155.08-149.96) | 131.10 (136.95-132.58) | 129.42 (134.88-130.57) | 173.52 (177.66-171.88) | 153.48 (158.20-153.08) | 160.98 (164.84-159.47) |
| (S1+S3)−>S2 | 169.50 (174.52-168.87) | 174.06 (178.64-172.84) | 151.86 (156.99-151.93) | 142.68 (148.70-143.95) | 153.42 (157.25-152.14) | 172.68 (177.52-171.76) |
| (S2+S3)−>S1 | 158.58 (163.33-158.05) | 172.26 (177.02-171.28) | 130.32 (134.13-129.79) | 151.20 (155.74-150.70) | 135.54 (140.39-135.88) | 151.62 (155.40-150.34) |
| S1−>S2 | 68.28 (70.86-68.58) | 80.64 (83.08-80.39) | 78.36 (81.32-78.71) | 74.76 (77.45-74.96) | 74.94 (77.02-74.53) | 71.82 (74.51-72.12) |
| S1−>S3 | 82.02 (85.42-82.69) | 77.10 (79.78-77.21) | 78.18 (81.43-78.83) | 73.20 (76.35-73.91) | 74.04 (76.38-73.91) | 81.96 (84.72-81.99) |
| S2−>S1 | 77.70 (80.44-77.85) | 64.62 (68.21-66.06) | 73.86 (76.40-73.94) | 63.36 (65.57-63.46) | 64.80 (66.58-64.42) | 67.92 (70.27-68.00) |
| S2−>S3 | 81.18 (84.22-81.51) | 74.40 (77.62-75.14) | 72.12 (74.58-72.17) | 74.34 (76.97-74.49) | 66.60 (68.60-66.38) | 64.68 (67.39-65.23) |
| S3−>S1 | 65.76 (68.20-66.01) | 73.02 (74.89-72.45) | 69.72 (72.49-70.17) | 65.88 (68.95-66.75) | 62.88 (64.39-62.30) | 64.02 (66.22-64.09) |
| S3−>S2 | 77.16 (79.75-77.18) | 73.38 (75.80-73.35) | 77.94 (80.40-77.81) | 65.52 (68.21-66.03) | 65.76 (67.24-65.05) | 65.88 (68.18-65.99) |
| **Average** | **103.73** | **102.29** | **95.75** | **98.27** | **94.61** | **100.17** |

Note: S1 means subject 1, S2 means subject 2, S3 means subject 3, CI means confidence level

**Table 11. Average training times of DSAN-MMD on CSI image dataset across all training percentages.**

| Task | Deep subdomain adaptation network (DSAN) with MMD loss | | | | | |
|---|---|---|---|---|---|---|
| | Training Time in seconds with different target data training samples percentages | | | | | |
| | 100% with (95% CI) | 80% with (95% CI) | 60% with (95% CI) | 40% with (95% CI) | 20% with (95% CI) | 10% with (95% CI) |
| (S1+S2)−>S3 | 136.32 (138.48-133.93) | 147.30 (152.75-147.84) | 150.42 (155.06-150.04) | 146.76 (151.96-147.06) | 152.16 (157.11-152.04) | 150.00 (154.52-149.52) |
| (S1+S3)−>S2 | 147.66 (152.65-147.72) | 136.56 (141.08-136.52) | 155.88 (160.52-155.32) | 134.04 (140.22-135.76) | 135.42 (139.82-135.31) | 156.36 (161.46-156.25) |
| (S2+S3)−>S1 | 131.22 (135.22-130.85) | 130.56 (135.16-130.81) | 128.88 (132.51-128.21) | 124.56 (129.15-125.00) | 141.96 (146.91-142.17) | 144.54 (149.22-144.41) |
| S1−>S2 | 70.32 (73.40-71.06) | 67.86 (70.65-68.39) | 74.22 (76.89-74.41) | 55.56 (57.56-55.71) | 63.78 (66.27-64.14) | 67.86 (70.91-68.65) |
| S1−>S3 | 58.32 (61.18-59.24) | 71.58 (74.26-71.88) | 65.28 (68.57-66.39) | 65.10 (68.35-66.18) | 76.50 (78.30-75.75) | 70.08 (72.84-70.51) |
| S2−>S1 | 56.04 (58.59-56.72) | 57.72 (61.32-59.40) | 63.42 (65.52-63.41) | 65.58 (67.81-65.63) | 55.86 (57.90-56.04) | 71.64 (74.32-71.93) |
| S2−>S3 | 59.94 (62.28-60.29) | 57.00 (59.76-57.86) | 60.96 (63.52-61.48) | 70.86 (73.14-70.78) | 59.58 (61.77-59.78) | 63.60 (66.68-64.56) |
| S3−>S1 | 60.90 (63.02-60.99) | 60.36 (62.27-60.26) | 58.68 (61.56-59.61) | 56.94 (59.90-58.00) | 56.58 (58.39-56.51) | 57.54 (59.99-58.07) |
| S3−>S2 | 57.78 (60.29-58.37) | 65.52 (68.17-65.98) | 61.68 (63.90-61.84) | 56.82 (59.64-57.75) | 57.84 (59.70-57.78) | 54.72 (56.69-54.87) |
| **Average** | **86.5** | **88.27** | **91.05** | **86.25** | **88.85** | **92.93** |

Note: S1 means subject 1, S2 means subject 2, S3 means subject 3, CI means confidence level

**Table 12. Average training times of DASAN-LMMD on CSI image dataset across all training percentages.**

| Task | Deep adversarial subdomain adaptation aetwork (DASAN) with LMMD loss | | | | | |
|---|---|---|---|---|---|---|
| | Training Time in seconds with different target data training samples percentages | | | | | |
| | 100% with (95% CI) | 80% with (95% CI) | 60% with (95% CI) | 40% with (95% CI) | 20% with (95% CI) | 10% with (95% CI) |
| (S1+S2)−>S3 | 133.20 (137.63-133.19) | 243.48 (246.60-238.48) | 228.00 (230.64-223.04) | 243.12 (244.92-236.81) | 233.34 (234.45-226.67) | 199.68 (202.58-195.92) |
| (S1+S3)−>S2 | 235.08 (237.55-229.71) | 224.28 (227.30-219.82) | 241.50 (244.19-236.14) | 245.16 (247.47-239.29) | 183.36 (186.66-180.55) | 228.30 (230.85-223.24) |
| (S2+S3)−>S1 | 224.64 (227.94-220.45) | 235.86 (239.21-231.35) | 240.96 (242.70-234.66) | 245.16 (246.65-238.48) | 216.60 (219.44-212.22) | 237.54 (237.95-230.03) |
| S1−>S2 | 120.06 (121.70-117.69) | 117.06 (118.88-114.98) | 122.34 (124.14-120.06) | 118.14 (119.37-115.43) | 105.90 (107.40-103.87) | 112.14 (113.68-109.94) |
| S1−>S3 | 137.82 (138.88-134.29) | 113.58 (115.06-111.27) | 140.16 (140.63-135.95) | 131.16 (132.15-127.78) | 98.52 (100.29-97.00) | 121.92 (123.39-119.33) |
| S2−>S1 | 133.38 (134.03-129.58) | 135.42 (136.12-131.60) | 114.84 (115.46-111.63) | 115.62 (116.81-112.95) | 97.80 (99.15-95.89) | 112.68 (114.25-110.50) |
| S2−>S3 | 130.56 (131.42-127.07) | 134.70 (135.64-131.15) | 127.26 (128.23-123.99) | 121.08 (122.41-118.38) | 108.24 (109.58-105.97) | 130.92 (131.49-127.13) |
| S3−>S1 | 114.24 (115.20-111.40) | 111.24 (112.76-109.05) | 105.00 (106.18-102.68) | 119.10 (120.48-116.51) | 89.10 (90.96-87.99) | 109.86 (110.75-107.09) |
| S3−>S2 | 120.84 (121.67-117.64) | 113.16 (114.03-110.26) | 110.82 (112.08-108.38) | 103.08 (104.76-101.33) | 89.10 (90.63-87.66) | 109.86 (110.96-107.30) |
| **Average** | **149.98** | **158.75** | **158.99** | **160.18** | **135.77** | **151.43** |

Note: S1 means subject 1, S2 means subject 2, S3 means subject 3, CI means confidence level

**Table 13. Average training times of DASAN-MMD on CSI image dataset across all training percentages.**

| Task | Deep adversarial subdomain adaptation network (DASAN) with MMD loss | | | | | |
|------|------|------|------|------|------|------|
| | Training Time (seconds) with different target data training samples percentages | | | | | |
| | 100% with (95% CI) | 80% with (95% CI) | 60% with (95% CI) | 40% with (95% CI) | 20% with (95% CI) | 10% with (95% CI) |
| (S1+S2)−>S3 | 193.74 (199.04-192.58) | 210.60 (212.76-205.74) | 195.00 (197.07-190.57) | 194.28 (196.58-190.11) | 170.16 (171.50-165.82) | 189.18 (192.03-185.72) |
| (S1+S3)−>S2 | 196.50 (198.50-191.95) | 206.46 (209.37-202.49) | 210.00 (212.67-205.67) | 195.72 (197.27-190.74) | 194.10 (197.81-191.34) | 200.10 (202.80-196.13) |
| (S2+S3)−>S1 | 212.34 (215.45-208.37) | 220.02 (223.15-215.82) | 193.02 (195.50-189.07) | 184.86 (186.40-180.24) | 196.20 (198.80-192.26) | 157.62 (158.41-153.15) |
| S1−>S2 | 111.78 (113.45-109.72) | 113.40 (115.16-111.38) | 116.22 (117.97-114.09) | 100.20 (101.36-98.02) | 99.24 (100.70-97.39) | 101.10 (102.34-98.97) |
| S1−>S3 | 104.46 (105.08-101.60) | 102.12 (103.43-100.02) | 98.34 (99.04-95.76) | 100.74 (101.48-98.12) | 103.08 (104.99-101.55) | 104.52 (105.60-102.11) |
| S2−>S1 | 95.82 (96.39-93.20) | 90.72 (90.63-87.61) | 85.98 (86.78-83.91) | 97.92 (99.07-95.80) | 88.68 (89.80-86.84) | 104.10 (105.53-102.06) |
| S2−>S3 | 101.04 (102.06-98.69) | 95.70 (95.85-92.66) | 101.22 (102.30-98.92) | 101.58 (102.58-99.19) | 97.62 (98.99-95.74) | 93.06 (93.62-90.52) |
| S3−>S1 | 88.08 (88.64-85.70) | 102.12 (103.51-100.11) | 84.36 (84.91-82.10) | 95.52 (96.10-92.92) | 98.34 (100.25-96.97) | 86.64 (87.41-84.52) |
| S3−>S2 | 89.94 (90.45-87.45) | 87.12 (87.75-84.85) | 94.20 (95.19-92.05) | 101.34 (102.95-99.57) | 90.24 (91.77-88.76) | 92.82 (93.93-90.84) |
| **Average** | **132.63** | **136.47** | **130.93** | **130.24** | **126.41** | **125.46** |

Note: S1 means subject 1, S2 means subject 2, S3 means subject 3, CI means confidence level

wireless signals. Even though sub-domain adaptation is usually considered to be a more significant method for cross-domain alignments because it fulfils the need of fine-grained information from the relevant classes of different domain, our simulations provide initial evidence that it is an inferior choice for human activity recognition (HAR) using device-free sensing.

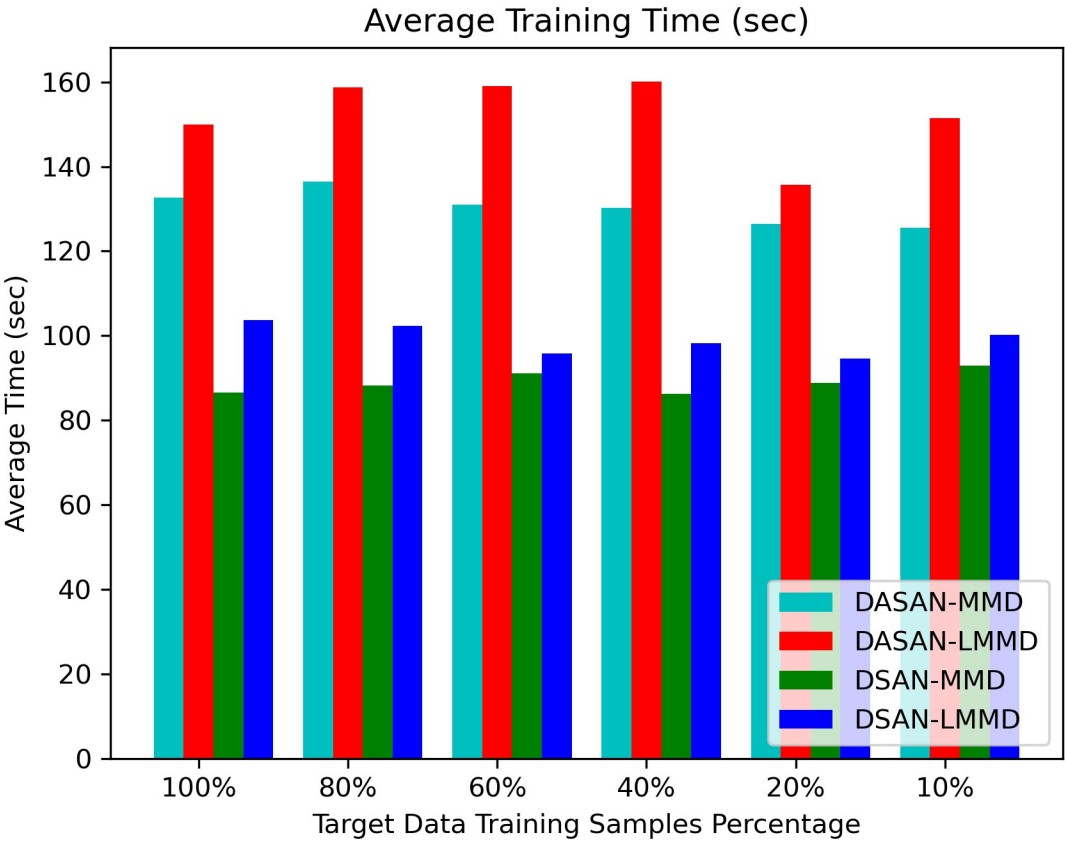

**Fig 9. Comparison of average training time of DASAN-MMD, DASAN-LMMD, DSAN-MMD, & DSAN-LMMD.**

Wireless signals show quite similar CSI patterns for different activities and it is not easy to align these sub-domains properly for better recognition accuracy. In other words, the distance between positive (samples belong to the same class) and negative (samples other than the positive class) samples is not reasonably large enough to align target sub-domains to their source counterparts using sub-domain alignment techniques such as LMMD. The adversarial AI model developed in this study shows improved predictive performance at all levels of test data proportion when compared to a non-adversarial model. We demonstrated the superiority of DASAN-MMD in terms of higher model recognition accuracy by comparing its transfer results with those of DASAN-LMMD, DSAN-LMMD, and DSAN-MMD. The experimental results further illustrate that we have developed a lightweight model with comparable simulation time to existing baseline methods. Our results show that MMD loss with an adversarial model aligns the source domain to the target domain globally, providing further evidence that inter-domain alignment is more effective for HAR using wireless signals and the dataset along with the preprocessing steps followed are suitable for such type of examinations.

## Author Contributions

**Conceptualization:** Muhammad Hassan.

**Formal analysis:** Muhammad Hassan, Fahrurrozi Rahman.

**Investigation:** Muhammad Hassan, Tom Kelsey, Fahrurrozi Rahman.

**Methodology:** Muhammad Hassan, Tom Kelsey.

**Software:** Muhammad Hassan, Fahrurrozi Rahman.

**Supervision:** Tom Kelsey.

**Validation:** Muhammad Hassan, Fahrurrozi Rahman.

**Writing – original draft:** Muhammad Hassan, Tom Kelsey.

**Writing – review & editing:** Muhammad Hassan, Tom Kelsey.

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
