## [Decision Letter · Decision Letter 0]

19 Nov 2023

PONE-D-23-32286Adversarial AI applied to cross-user inter-domain and intra-domain adaptation in human activity recognition using wireless signalsPLOS ONE

Dear Dr. Kelsey,

Thank you for submitting your manuscript to PLOS ONE. After careful consideration, we feel that it has merit but does not fully meet PLOS ONE’s publication criteria as it currently stands. Therefore, we invite you to submit a revised version of the manuscript that addresses the points raised during the review process.

We look forward to receiving your revised manuscript.

Kind regards,

Lei Chu

Academic Editor

PLOS ONE

Journal Requirements:

3. We note you have included a table to which you do not refer in the text of your manuscript. Please ensure that you refer to Table 1-10 in your text; if accepted, production will need this reference to link the reader to the Table

Additional Editor Comments (if provided):

Thank you for submitting your manuscript. We have now completed the review process, and I am writing to inform you of the decision regarding your submission.

After careful consideration and a thorough evaluation by our peer reviewers, we have decided that your manuscript can only be accepted for publication following major revisions. The reviewers have raised several important points that need to be addressed before we can move forward.

We appreciate the opportunity to consider your work and believe that these revisions will significantly improve the manuscript’s impact and clarity. If you have any questions or need further clarification regarding the reviewers' comments or the revision process, please do not hesitate to contact us. Thank you for considering PLOS ONE for your work. We look forward to receiving your revised manuscript.

Reviewers' comments:

Reviewer's Responses to Questions

**Comments to the Author**

1. Is the manuscript technically sound, and do the data support the conclusions?

Reviewer #1: Yes

Reviewer #2: Yes

2. Has the statistical analysis been performed appropriately and rigorously? 

Reviewer #1: Yes

Reviewer #2: Yes

3. Have the authors made all data underlying the findings in their manuscript fully available?

Reviewer #1: Yes

Reviewer #2: Yes

4. Is the manuscript presented in an intelligible fashion and written in standard English?

Reviewer #1: Yes

Reviewer #2: Yes

5. Review Comments to the Author

Reviewer #1: In this paper, adversarial AI has been applied to cross-user inter-domain and intra-domain adaptation in human activity recognition using wireless signals. The proposed idea is new and interesting. The authors also conducted extensive experiments. I can recommend this paper for a publication. However, there are several main concerns that need to be addressed properly.

1.The paper's abstract needs further revision. The authors should shorten the abstract in a precise way.

2.The main contributions should be summarized one by one in the introduction.

3.In related works, the authors summarize previous literatures instead of only listing them. On this basis, the novelty should be highlighted by presenting what has been previously explored or not.

4.The authors should place the related work section after the introduction section.

5.In particular, I strongly recommend the authors to restructure the tables and enlarge the font size, so as to make them more clear.

6.I also recommend the authors to refer to related activity recognition literatures:10.1016/j.knosys.2023.110789; 10.1109/TKDE.2023.3277839; 10.1109/JBHI.2023.3275438; 10.1109/TII.2023.3315773.

Reviewer #2: Adversarial AI applied to cross-user inter-domain and intra-domain adaptation in

human activity recognition using wireless signals – Comments

This submission proposes a deep learning framework that deals with the inter-domain and intra-domain adaptation problem in human activity recognition tasks. The consideration of WiFi CSI as the input is innovative and the use of adversarial AI for cross-user inter-domain and intra-domain adaptation with wireless signals addresses some significant challenges in the HAR field. However, from the proposed method and the experimental verifications of this paper, I do not think it can bring enough improvements to this field.

1.While the introduction part presents basic background information and concepts relevant to this study, a brief summary of the primary contributions at the outset would help understand the scholar value and academic impact of the research.

2.The problem definition section of the manuscript illustrates the challenges inherent in this research area, but the formulation of these issues lacks precision and clarity. A clearer definition of the problem is needed to be provided for subsequent proposed methods and experiments.

3.For the proposed adversarial AI model, firstly, the theoretical foundations of the model are not well developed and a more robust theoretical framework is needed to support the integrity of the model. Also, there is a lack of innovative approaches and unique contributions that are currently existing in the field of adversarial AI. While the network structure employed in the model is common for a variety of inputs such as images, it would be better to have explanations on specific designs for handling WiFi CSI inputs.

4.The figures used in this paper can be better structured and illustrated.

In conclusion, I do not think this article can be accepted at this time.

6. PLOS authors have the option to publish the peer review history of their article (what does this mean?). If published, this will include your full peer review and any attached files.

Reviewer #1: No

Reviewer #2: No

---

## [Author Response · Author response to Decision Letter 0]

3 Jan 2024

Thanks for your decision email. This letter details our revisions to the manuscript that take into account the reviewers’ comments. The comments are quoted in full; our comments follow each point raised.

Journal Requirements:

Changes have been made in the previously submitted document regarding PLOS ONE's style requirements. Please see the track changes.

- Please note that PLOS ONE has specific guidelines on code sharing for submissions in which author-generated code underpins the findings in the manuscript. In these cases, all author-generated code must be made available without restrictions upon publication of the work. Please review our guidelines at https://journals.plos.org/plosone/s/materials-and-software-sharing#loc-sharing-code and ensure that your code is shared in a way that follows best practice and facilitates reproducibility and reuse.

Code for the proposed model is available in a GitHub repository in open access at: https://github.com/DASAN-MMD/Code. A link to the repository is provided in the data and code availability section of the revised version.

- We note you have included a table to which you do not refer in the text of your manuscript. Please ensure that you refer to Table 1-10 in your text; if accepted, production will need this reference to link the reader to the Table

Many thanks for reminding us about this. All the tables have been referred in the text in the revised version. Please see the track changes. 

Reviewer #1: 

- The paper's abstract needs further revision. The authors should shorten the abstract in a precise way.

There are many changes and removal of content from abstract to summarise it properly. Please see the revised version.

- The main contributions should be summarized one by one in the introduction.

Thanks for your suggestion. Our main contributions have been summarised in the end of the introduction. Please see the revised version.

- In related works, the authors summarize previous literatures instead of only listing them. On this basis, the novelty should be highlighted by presenting what has been previously explored or not.

Many thanks for pointing out this area. We have discarded the irrelevant explanation and listed the previous work in the revised version.

- The authors should place the related work section after the introduction section.

Changes have been made as per the suggestion.

- In particular, I strongly recommend the authors to restructure the tables and enlarge the font size, so as to make them more clear.

Thanks for your recommendation. It was difficult to enlarge the font size in previous submission as the tables were going out of the page margin. To do that, we have split each of the old tables into two and enlarged their font.

- I also recommend the authors to refer to related activity recognition literatures:10.1016/j.knosys.2023.110789; 10.1109/TKDE.2023.3277839; 10.1109/JBHI.2023.3275438; 10.1109/TII.2023.3315773.

Thanks for referring to some of the valuable content in the similar area. We went through this content, in fact some of them are cited in our related work session.

Reviewer #2: 

- While the introduction part presents basic background information and concepts relevant to this study, a brief summary of the primary contributions at the outset would help understand the scholar value and academic impact of the research.

Thanks for your suggestion. Our main contributions have been summarised in the end of the introduction. Please see the revised version.

- The problem definition section of the manuscript illustrates the challenges inherent in this research area, but the formulation of these issues lacks precision and clarity. A clearer definition of the problem is needed to be provided for subsequent proposed methods and experiments.

Thanks for pointing out this. Though, in previous submission we tried to make the problem statement very simple for the reader, but we agree it’s a bit vague. We have made several changes in the problem definition in our revised version. Please see the track changes.

- For the proposed adversarial AI model, firstly, the theoretical foundations of the model are not well developed and a more robust theoretical framework is needed to support the integrity of the model. Also, there is a lack of innovative approaches and unique contributions that are currently existing in the field of adversarial AI. While the network structure employed in the model is common for a variety of inputs such as images, it would be better to have explanations on specific designs for handling WiFi CSI inputs.

Thanks for your comments. Additional text has been added related to specific designs for handling WiFi CSI inputs. Each module of the model has also been elaborated with mathematical expressions to provide more clarity. We completely agree with the statement that the proposed model is very common for a variety of image inputs but it’s a robust and simple model suitable for the analysis of global and sub-domain alignment using wireless sensing, which is our key contribution and its acceptability on CSI Image data for cross-user domain shifting task is a new initiative in this field. We have also shown key findings on a public CSI dataset using specific pre-processing steps. In summary, our model is not going to revolutionize this entire field, but it opens door for many key challenges in this field as future directions, such as investigating combined cross-environmental and cross-user features to make the WiFi sensing more appealing and applicable.

- The figures used in this paper can be better structured and illustrated.

We have improved figure structure. See the revised version.

---

## [Decision Letter · Decision Letter 1]

1 Feb 2024

Adversarial AI applied to cross-user inter-domain and intra-domain adaptation in human activity recognition using wireless signals

PONE-D-23-32286R1

Dear Dr. Kelsey,

We’re pleased to inform you that your manuscript has been judged scientifically suitable for publication and will be formally accepted for publication once it meets all outstanding technical requirements.

Kind regards,

Sunder Ali Khowaja, Ph.D.

Academic Editor

PLOS ONE

Additional Editor Comments (optional):

The reviewers have submitted the reports. The reviewers are satisfied with the revisions. However, one of the reviewers have minor concerns, which is the description of the work being done should be improved. Authors should incorporate the comments while submitting the final version of the paper.

Reviewers' comments:

Reviewer's Responses to Questions

**Comments to the Author**

1. If the authors have adequately addressed your comments raised in a previous round of review and you feel that this manuscript is now acceptable for publication, you may indicate that here to bypass the “Comments to the Author” section, enter your conflict of interest statement in the “Confidential to Editor” section, and submit your "Accept" recommendation.

Reviewer #1: All comments have been addressed

Reviewer #2: All comments have been addressed

2. Is the manuscript technically sound, and do the data support the conclusions?

Reviewer #1: Yes

Reviewer #2: Yes

3. Has the statistical analysis been performed appropriately and rigorously? 

Reviewer #1: Yes

Reviewer #2: Yes

4. Have the authors made all data underlying the findings in their manuscript fully available?

Reviewer #1: Yes

Reviewer #2: Yes

5. Is the manuscript presented in an intelligible fashion and written in standard English?

Reviewer #1: Yes

Reviewer #2: Yes

6. Review Comments to the Author

Reviewer #1: The authors have addressed my concerns satisfactorily, and I can recommend this paper for a publication.

Reviewer #2: The experimental analysis of the article is relatively comprehensive, and the relevant code and data has open source. However, the description of the article needs to be carefully checked. For example, please check the description in contribution for errors.

7. PLOS authors have the option to publish the peer review history of their article (what does this mean?). If published, this will include your full peer review and any attached files.

Reviewer #1: No

Reviewer #2: No

---

## [Editor Report · Acceptance letter]

15 Feb 2024

PONE-D-23-32286R1 

PLOS ONE

Dear Dr. Kelsey, 

I'm pleased to inform you that your manuscript has been deemed suitable for publication in PLOS ONE. Congratulations! Your manuscript is now being handed over to our production team.

Kind regards, 

on behalf of

Dr. Sunder Ali Khowaja 

Academic Editor

PLOS ONE